# The Development and Effectiveness of a Clinical Training Violence Prevention Program for Nursing Students

**DOI:** 10.3390/ijerph17114004

**Published:** 2020-06-04

**Authors:** Yunhwa Jeong, Kyunghee Lee

**Affiliations:** 1Department of Nursing and Kyongbuk Science College, Gyeongsangbuk-do 39913, Korea; jyw2430@hanmail.net; 2College of Nursing, Keimyung University, Daegu 42601, Korea

**Keywords:** workplace violence, nursing students, coping style, prevention program

## Abstract

The study aimed to develop and evaluate a violence prevention program for nursing students to improve communication self-efficacy, problem-focused coping style, emotion-focused coping style, and the ability to cope with violence. Using an eight-session violence prevention program, the study was designed as quasi experimental, with a pretest, posttest, and follow-up assessment with a nonequivalent control group. Nursing students from the fourth year of a university were selected as participants; 22 students were assigned to the experimental group and 23 to the control group. Data analysis included Chi-square, Fisher’s exact test, Levene’s Test, Mann–Whitney U-test, and repeated measures ANOVA. Results showed that the experimental group showed significantly higher posttest scores for the problem-focused coping style (F = 20.77, *p* < 0.001), intra-individual and interaction effects for the emotion-focused coping style (F = 12.03, *p* < 0.001), and the ability to cope with violence (U = 70, *p* < 0.001) than the control group. Thus, the workplace violence prevention program was effective for nursing students.

## 1. Introduction

### 1.1. Need for Study

Many types of workplace violence occur in the healthcare field [1,2,3], including physical violence, injury, blackmail, insult, defamation, interference with work, and indecent assault [4,5]. Specifically, nurses are exposed to risks of physical and psychological violence; violence to nurses in healthcare centers in South Korea and abroad has recently been recognized as a social problem [6,7,8,9,10,11,12].

In the USA, surveys of 3765 registered nurses highlighted the prevalence of workplace violence among nurses, that 21% registered nurses and nursing students reported being physically assaulted in a 12-month period, and that more than half of the nurses and nursing students were verbally insulted [13,14].

The majority of nursing students at clinical training sites experienced diverse types of violence, including from patients, caregivers, nurses, doctors, and hospital employees. They experienced violence in emergency rooms, general wards, psychiatric wards, and intensive care units; as many as 40% to 98.2% of study respondents experienced violence [6,15,16,17,18]. Nursing students experiencing verbal abuse or physical violence complained of psychological problems, such as anxiety, fear, decreased self-confidence, anger, and hostility, as well as physical problems caused by trauma; some of them even wanted to quit their nursing program [6,18,19,20,21,22]. Many used passive emotion-focused coping styles of neglect or avoidance instead of asking for help or reporting violence [17,22,23,24].

Coping is defined as a process and method of controlling stress with cognitive behavior, by attempting to actively regulate internal and external demands. A coping style is defined as the specific process and method of dealing with stress and emotions. Problem-focused coping includes coping styles such as collecting information about the problem and making a behavioral plan to directly address or alter the stressful event; emotion-focused coping includes coping styles such as maintaining distance, self-condemnation, and imagination to relieve negative emotional distress related to a stressful situation [25].

Research has shown that nursing students are faced with serious problems caused by violence during clinical training in hospitals, and this adversely affects their roles and work performance. It is, therefore, necessary to develop a violence prevention program to provide knowledge and promote self-confidence so that nursing students can cope with and manage any inappropriate, threatening, or violent event during clinical training and develop a situational intervention [18,26]. As a strategy of dealing with violent behavior, training to reinforce communicative competence and coping ability has been introduced [27,28,29,30,31,32,33]. The healthcare field, and nursing in particular, requires an effective strategy to reduce workplace violence at an individual level [1,34]. To do so, a specialized counseling system, relevant laws, or a center focused on specific solutions may be implemented at organizational or national levels [7,35,36,37].

Limited domestic and international studies have applied violence prevention programs to nursing students that addressed coping with and managing violence, prevention training [30,37,38,39,40,41,42,43,44], online violence prevention and coping education [45], etc. However, it is important that every nursing college and hospital implement a regular violence prevention program to help nursing students cope with violence from patients, caregivers, and staff; systematic education should be implemented prior to clinical training with the aim of protecting nursing students from violence at clinical training sites [17,22,23,24,46,47].

Violence prevention programs should address how to improve self-confidence in communication during a violent situation [32,41] and styles of coping with violence [23]. Discussion and role play through lecture-based education and group activity have been found to be effective in such programs [22,39,46,47,48,49,50].

Research has shown that a teaching method involving a real-life violent situation and simulation where the participants were asked to prevent and cope with violence was very effective in developing nursing students’ coping abilities with violent situations during clinical training [42]. Communication skills and the art of self-defense are two basic styles of coping with violent situations; however, few violence prevention programs have contained both styles [51,52,53].

Accordingly, this study aimed to develop and evaluate a violence prevention program for nursing students in clinical training that incorporated communication skills and the art of self-defense; to do so, a conceptual model was developed based on the programmed conceptual model of Kim [54], as applied to nursing students. This study intended to determine the ultimate effects of this program on communication self-efficacy, problem-focused coping style, and emotion-focused coping style in nursing students and to evaluate the violent situation simulation program in terms of nursing students’ ability to cope with violence in practice.

### 1.2. Purpose

This study aimed to develop and apply a violence prevention program for nursing students in clinical training and evaluate its effects.

### 1.3. Hypotheses

**Hypotheses 1** **(H1).**
*The experimental group participating in a violence prevention program has a higher level of communication self-efficacy than the control group.*


**Hypotheses 2** **(H2).**
*The experimental group participating in a violence prevention program has a higher level of problem-focused coping style than the control group.*


**Hypotheses 3** **(H3).**
*The experimental group participating in a violence prevention program has a lower level of emotion-focused coping style than the control group.*


**Hypotheses 4** **(H4).**
*The experimental group participating in a violence prevention program scores higher in observational evaluation of the ability to cope with violence than the control group.*


### 1.4. Research Program Development Steps

Based on the model by Kim [54], which is shown in Figure 1, a conceptual model of this program was constructed and is shown in Figure 2. 

In this study, Step 1 was planning, Step 2 composition, Step 3 implementation, and Step 4 evaluation. Step 1 involved setting a program goal and conducting a literature review and needs analysis. Step 2 involved developing program intervention techniques, setting length and timing of sessions, grouping, composing program contents, and conducting preliminary research. During Step 3, the violence prevention program was implemented. Step 4—evaluation—involved measuring communication self-efficacy and problem- and emotion-focused coping styles, simulating violent situations in emergency rooms, and rating the ability to cope with violence through observational evaluation.

## 2. Materials and Methods

### 2.1. Study Design

This is quasi-experimental research using a nonequivalent control group pretest-posttest design to develop and evaluate a violence prevention program for effectiveness.

### 2.2. Participants

The researchers recruited nursing students who were seniors in the Nursing Department at K University in North Gyeongsang Province in South Korea. Eligible participants were students who had completed at least one year of clinical training and who had experienced verbal abuse, physical threats, physical violence, or sexual violence from patients, caregivers, nurses, doctors, or other staff members at clinical training centers at least once during the clinical training period. To ensure that informed consent was obtained, students were informed of the purpose of this study, and then they provided written consent. Students selected as participants were randomly assigned to the experimental group (*n* = 22) and the control group (*n* = 23), according to the training cycle. All the participants were female and in their early 20s. The mean age of the experimental group was 22.4 years and that of the control group was 24 years. To determine the sample size, G*Power 3.1.2 [55] was used with a testability (1-β) of 0.95, a significance level (α) of 0.05, and an effect size (f) of 0.25; the goal sample size was 44 students. Taking into account the dropout rate, 26 students were chosen for each of the experimental and control groups. After excluding four students from the experimental group and three from the control group based on a lack of participation due to personal reasons, 45 students were finally included.

### 2.3. Instruments

#### 2.3.1. Communication Self-Efficacy

The Counseling Self-Estimate (COSE) Inventory, which was developed by Larson [56], translated by Hong [57], and adapted by Park [58], was used. A higher score means a higher level of communication self-efficacy. In terms of reliability, Cronbach’s α was 0.74 in Park [58] and 0.76 in this study. The questionnaire measures communication self-efficacy with a total of 37 items in four areas: communication skills (12 items), counseling process (10 items), dealing with difficult patient behavior (7 items), ability to deal with cultural differences (4 items), and awareness of values (4 items). The responses are based on a six-point Likert scale, with higher scores indicating higher levels of communication self-efficacy.

#### 2.3.2. Coping Style

Based on the theory of Lazarus and Folkman [25], Bae [59] created an instrument of 16 items to measure problem-focused and emotion-focused coping styles. The instrument was additionally adapted for this study by adding two items related to the art of self-defense from the Self-Confidence in Coping with Patients’ Assault Scale developed by Thackrey [60]. The revised scale had 18 items, and the content validity index was ≥0.90 after content validation by an expert. Cronbach’s α was 0.71 in Bae [59] and 0.74 in this study. In this study, 18 items from the instrument were used to evaluate coping style, with responses on a five-point Likert scale. It consists of problem-focused coping style (11 items) and emotion-focused coping style (7 items).

#### 2.3.3. Ability to Cope with Violence

Ten items related to violence risk and safety management, communication, and attitude from the sub-items of the instrument developed by Seo [61] were used for an observational assessment. The following scores were assigned: score 0—not performed, 1—poor, and 2—performed. In terms of reliability, Cronbach’s α was 0.97 in Seo [61] and 0.88 in this study. The questionnaire consisted of 10 questions, a scale of 3 points ranging from 0–2 (items 3 and 8), and a scale of 2 points from 0–1 (items 1,2,4,5,6,7,9 and 10).

### 2.4. Data Collection

This study was approved by the Institutional Review Board (IRB NO. 40525-201711-HR-78-05) of K University. Before beginning the research, the researcher personally clarified the purpose, contents, and methods of the research with the dean of the faculty for the Nursing Department at K University and obtained permission to conduct the research. Participants’ general characteristics, communication self-efficacy, and problem-focused and emotion-focused coping styles were measured in both the experimental and control groups. Posttest was performed in both groups after the end of the eight-session program, and three professors performed an observational assessment of the participants’ ability to cope with violence while running the simulation. To minimize inter-rater measurement errors, protocols regarding data collection methods, procedures, and tools were created to train the assistants.

Three observers performed an observational assessment of the participants’ ability to cope while running the simulation. As for internal consistency among the three observers, Cronbach’s α was estimated at ≥0.9. Participants in the experimental group were paid 20,000 won for participation in each session of the program and 5000 won for a meal; participants in the control group were paid 15,000 won for participation in the research. Refreshments were also provided during the simulation.

### 2.5. Researchers’ Preparation for Program Development

The principal researcher and two research assistants participated in the study. The first researcher is a professor of psychiatric nursing and has 17 years of clinical experience and 10 years of educational experience. She has extensive experience in communication training programs for nursing students. The second researcher is also a professor of psychiatric nursing and has more than 30 years of educational experience. The first research assistant is a professor of adult nursing and has been running a program for simulation of adult nursing for students for 3 years. The second research assistant is a professor of adult nursing and has 3 years of experience in basic nursing simulation and adult nursing simulation teaching for nursing students. In the simulation, three professional actors were employed as research assistants. The first researcher explained the purpose of the violence prevention program to two professional lecturers before the program. The key concepts, activities, and contents were also explained and shared to proceed with the program 

### 2.6. Development of Violence Prevention Program

To develop a violence prevention program, intervention program techniques, length, timing of sessions, grouping, and theoretical frameworks were analyzed through a literature review based on the program development model suggested by Kim [55]. The feasibility and applicability of preliminary research was assessed for applicability to this program; an expert validated the instruments prior to the preliminary research.

Before designing the program, a literature review was performed to select the theme and goal of each session. To develop the program’s content, previous studies regarding violence experienced by nursing students, violence at hospitals, and violence prevention programs for nursing students and nurses at the national and international levels were reviewed.

Two researchers and a special lecturer who served as an assistant conducted the sessions, and three professional performers were employed to act as patients in the simulation. For two days, the preliminary program was run as an intensive. The students suggested the need for a full explanation of the definition and contents of violence at the introduction stage. They also indicated that their personal participation would be less boring and attract greater attention than a lecture-based program. Their opinions were incorporated by adding discussion, interview skills training based on self-examination of communication, role play, and the art of self-defense in a 1:1 real-life situation to the program. The violent situation simulation involved verbal abuse and physical violence by an acute pancreatitis patient who visited an emergency room with upper abdominal pain complaints; the patient then became mad when a nurse requested reception. The simulation was revised and complemented by an expert. The finally confirmed program is shown in Table 1.

The violence prevention program was conducted for the students in the experimental group in a total of eight sessions—120 min per session, two sessions per week—for four weeks from 1 May to 5 June 2019. The students in the control group were given a 120-min lecture in the first session.

Session 1 involved an introduction to the program introduction, explanation of group rules, videos on hospital violence, development of solutions, and presentation of suggestions. Session 2 involved writing a scenario and role play presentation after watching videos as well as discussing therapeutic and non-therapeutic communication. Session 3 involved listening and good speech training. Session 4 involved 1:1 interview training for participants to examine their own communication, detect their own problems, engage in introspection, and incorporate changes to their therapeutic communication. Session 5 involved watching YouTube videos related to verbal abuse, followed by composing verbal abuse cases, writing a scenario, and a role play, as an attempt to prevent and cope with verbal abuse. Session 6 involved watching YouTube videos related to physical violence, followed by writing a scenario, and a role play using therapeutic and non-therapeutic communications. Session 7 involved watching YouTube videos related to physical violence, followed by composing physical violence cases, writing a scenario, and a role play, as an attempt to prevent and cope with physical violence; and Session 8 involved consolidating violence prevention and coping methods, issuing a certificate of completion, completing a posttest, and evaluating educational satisfaction to understand the effectiveness of the program. Three months after completion of the program, the simulation was conducted for the observational assessment to determine the persistence of the effectiveness. Watching YouTube videos, writing a scenario, role play presentation, listening training, good speech training, assertiveness training, and the art of self-defense were used to promote the goal of each stage in the conceptual model of the violence prevention program. The program assessment involved pretest, posttest, and follow-up assessment.

### 2.7. Data Analysis

The collected data were processed by using SPSS/WIN 25.0 to perform frequency analysis and multiple-answer analysis for the participants’ general characteristics and experiences with violence; *x*^2^ test and Fisher’s exact test were performed to determine homogeneity. Repeated measures ANOVA was performed to analyze the variables of communication self-efficacy and problem-focused and emotion-focused coping styles for inter-group differences, degree of intra-group changes, and interaction effects. The non-parametric method of the Mann–Whitney U-test was used to compare mean differences in the ability to cope with violence between the experimental and control groups.

## 3. Results

### 3.1. General Characteristics and Homogeneity Test

The participants’ general characteristics are presented in Table 2. Among the participants, 68.9% experienced a violent situation during a day shift, 71.1% reported that there was no security in the clinical setting, and 62.2% suggested the urgent need for relevant education. Prior to the program, the *x*^2^ test and Fisher’s exact test found no significant inter-group differences between the experimental and control groups, which indicated homogeneity in the general characteristics.

### 3.2. Homogeneity Test for Dependent Variables

Homogeneity tests were performed for dependent variables such as communication self-efficacy, problem-focused and emotion-focused coping styles, and the ability to cope with violence between the experimental and control groups prior to the program. Communication self-efficacy and problem-focused and emotion-focused coping styles were homogeneous, whereas the ability to cope with violence was not homogeneous (Table 3).

### 3.3. Box’s Test for Equality of Covariance Matrix

Homogeneity for repeated measures ANOVA was tested using Box’s test for equality of covariance matrix. Communication self-efficacy, problem-focused coping style, and emotion-focused coping style were found to satisfy the assumption of homogeneity of variance (*p* > 0.05) (Table 4).

Analysis of the normal distribution of the measured variables with the Kolmogorov–Smirnov test in the experimental group and the control group revealed that the ability to cope with violence was not normally distributed and was analyzed with the Mann–Whitney U-test.

The sphericity assumption indicates the equivariance condition in which the correlation between each pair of data measured repeatedly is the same, and that the variances in each process are the same for each process. If the sphericity assumption is satisfied, univariate analysis is used (Table 5).

Based on the result of Mauchly’s sphericity test, communication self-efficacy, problem-centered coping style, and emotion-centered coping style satisfied the assumption (*p* > 0.05), so univariate analysis was used.

### 3.4. Hypothesis Testing for Effectiveness of Violence Prevention Program

#### 3.4.1. Testing of Hypothesis 1

A repeated measures ANOVA was used to test Hypothesis 1—the experimental group participating in a violence prevention program has a higher level of communication self-efficacy than the control group. Results showed no statistically significant inter-group changes in communication self-efficacy (F = 0.03, *p* = 0.845), time points (F = 1.08, *p* = 0.343), or period-group interaction (F = 0.83, *p* = 0.437). Thus, Hypothesis 1 was rejected (Table 6).

#### 3.4.2. Testing of Hypothesis 2

A repeated measures ANOVA was used to test Hypothesis 2—the experimental group participating in a violence prevention program has a higher level of problem-focused coping style than the control group. Results showed that statistically significant inter-group changes in problem-focused coping style (F = 31.61, *p* < 0.001), time points (F = 47.66, *p* < 0.001), and period-group interaction (F = 20.77, *p* < 0.001). Thus, Hypothesis 2 was supported (Table 6).

#### 3.4.3. Testing of Hypothesis 3

A repeated measures ANOVA was used to test Hypothesis 3—the experimental group participating in a violence prevention program has a lower level of emotion-focused coping style than the control group. Results no statistically significant inter-group changes in emotion-focused coping style (F = 2.08, *p* = 0.156) but statistically significant changes in time points (F = 12.35, *p* < 0.001) and period-group interaction (F = 12.03, *p* < 0.001). Thus, Hypothesis 3 was partially supported (Table 6).

#### 3.4.4. Testing of Hypothesis 4

The observational assessment was used to test Hypothesis 4—the experimental group participating in a violence prevention program scores higher in observational evaluation of the ability to cope with violence than the control group. Results showed that the experimental group scored statistically significantly higher for the ability to cope with violence than the control group (U = 70, *p* < 0.001). Thus, Hypothesis 4 was supported (Table 7).

#### 3.4.5. Post-Test Comparing Groups between Time Points: Multiple Comparison Analysis

If the interaction between time and group is determined through repeated measurement and variance analysis and if the change over time between the two groups is different, we should check at what point the difference occurs at a later time. After adjusting the significance level through Bonferroni’s method, the two groups can be compared for each time point with an independent sample t-test. Communication self-efficacy did not require a post-test because the interaction was not significant (Figure 3). 

In problem-focused coping style, if the change over time between the two groups is different, we should check at what point the difference occurs through a post-test. The corrected significance level is 0.05/3 = 0.017 because pre, post, and follow-up are performed to correct the significance level through a post-test with Bonferroni’s method that compares groups by time point. Problem-focused coping style can be said to be different between groups in post and follow-up (Figure 4).

In emotion-focused coping style, if the change over time between the two groups is different, we should check at what point the difference occurs through a post-test. The corrected significance level is 0.05/3 = 0.017 because pre, post, and follow-up are performed to correct the significance level through a post-test with Bonferroni’s method comparing groups between time points. The emotion-focused coping style can be said to be different between groups in follow-up (Figure 5).

## 4. Discussion

This study aimed to determine the effects of a violence prevention program on communication self-efficacy, problem-focused and emotion-focused coping styles, and the ability to cope with violence in nursing students. In addition, it aimed to provide supporting data that could help improve nursing students’ ability to prevent and cope with violence in nursing education.

As a result of the program, communication self-efficacy slightly increased, without statistically significant effects. This result is similar to the finding that case-by-case SBAR (Situation-Background-Assessment-Recommendation) led to a statistically insignificant improvement in communication self-efficacy through a communication training program due to clinical knowledge of cases and the lack of self-confidence among nursing students who were sophomores [62]. In addition, this result is supported by a study that found that the more experienced nurses were in managing violence in practice, the higher their level of self-efficacy [63]. Students in this study severely lacked nursing experience with patients as well as experience with violence; despite the slight improvement in communication self-efficacy, the follow-up assessment found no continuous, significant improvement in communication self-efficacy. This is likely because the follow-up assessment was conducted three months after the end of the eight-session program, without any intervening education. While the communication training involved self-examination and responding to complaints during the interview, the students seemed to have difficulty with these tasks, which may have affected the results. The slight improvement in communication self-efficacy after the program is likely due to the therapeutic and non-therapeutic communication training and role-playing exercises. Education utilizing simulated patients that help students feel more self-confident as well as continuous, regular communication training seems to be necessary to improve communication self-efficacy [30,64].

The experimental group experienced more statistically significant effects in terms of the problem-focused coping style in the posttest, and follow-up assessment, than the control group. This result is similar to the finding of Park, Kim [6] that nurses asked for help from their colleagues and made a verbal report to their supervisor immediately after experiencing violence; it is also supported by another finding that training involving a scenario of a violent encounter improved self-defense and increased self-confidence [37,38]. In this program, communication training using real-life role play or simulation likely prepared students to better cope with violence using a problem-focused style.

Similarly, the experimental group showed more statistically significant effects in terms of emotion-focused coping styles in the follow-up assessment than the control group. This result is supported by another study’s finding that a psychological education program led to a reduction in coping based on avoidance [65]. In this study, the impact on emotion-focused coping is likely because students engaged in role play and developed a problem-solving ability to cope immediately with any violent event based on emotional detachment, instead of avoidance. When experiencing violence, participants in the experimental group had better coping abilities and a better awareness of violence; they made a report of a violent event to a nurse or nurse manager, instead of neglecting or enduring it. This result is supported by the suggestion that a workplace violence prevention and management program should include a focus on communication skills [32,36,53].

The observational assessment for the ability to cope with violence was performed through a simulation at the evaluation stage. This study is differentiated from the previous research of Choi and Lee [66], and Choi, Cho, Cho and Kim [67], in that it conducted the simulation three months after the end of the program and determined continued effectiveness of the program. During the simulation, participants viewed videos related to the art of self-defense and demonstrated it personally in a 1:1 team, employing therapeutic communication skills and the art of self-defense to cope immediately with violence until security arrives. The experimental group scored significantly higher in the observational assessment for the ability to cope with violence than the control group. In particular, students in the experimental group explained that they were able to immediately apply the art of self-defense in the simulation because of personal practice. Based on these results, this violence prevention program seems to be effective in helping nursing students improve the problem-focused coping style. In conclusion, practice in coping with a violent situation through role play and simulation three months after the end of the program along with observational assessment of the ability to cope with violence seemed to improve the ability to cope with violence.

The limitation of this study is that students who apply for the program in the 4th grade are unable to see the long-term effects and changes in coping with workplace violence incidents by employing students as a nurse.

## 5. Conclusions

This study developed and applied a violence prevention program for nursing students and confirmed that the program was an effective intervention for improving the problem-focused coping style, improving the ability to cope with violence, and reducing the emotion-focused coping style. The investigation of the behavioral changes after program intervention revealed that the scores of coping with violence were higher in the experimental group than in the control group. The violence prevention program for nursing college students in Korea is not implemented currently, but if the program developed in this study would be standardized and applied as a regular education before the clinical practicum, it would be helpful to prevent the nursing students from violence during the clinical practicum. Thus, it is recommended that nursing students be provided with systematic education in nursing schools and training centers in the form of a violence prevention program prior to participating in clinical training.

## Figures and Tables

**Figure 1 ijerph-17-04004-f001:**
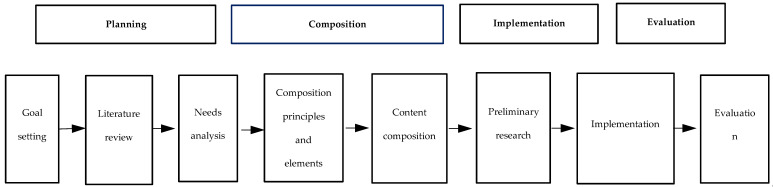
Kim’s (2002) conceptual framework.

**Figure 2 ijerph-17-04004-f002:**
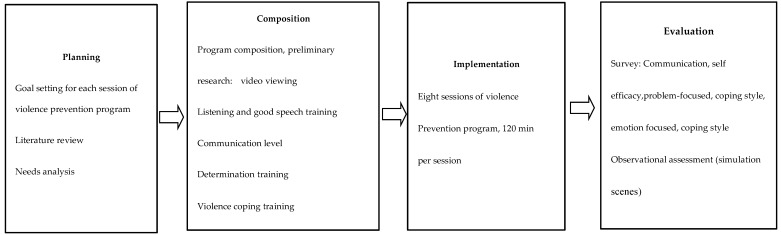
Conceptual framework of this study.

**Figure 3 ijerph-17-04004-f003:**
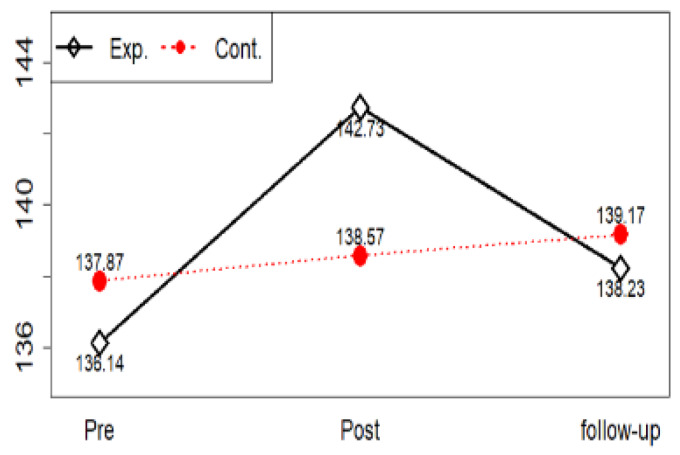
Communication self-efficacy profile plots.

**Figure 4 ijerph-17-04004-f004:**
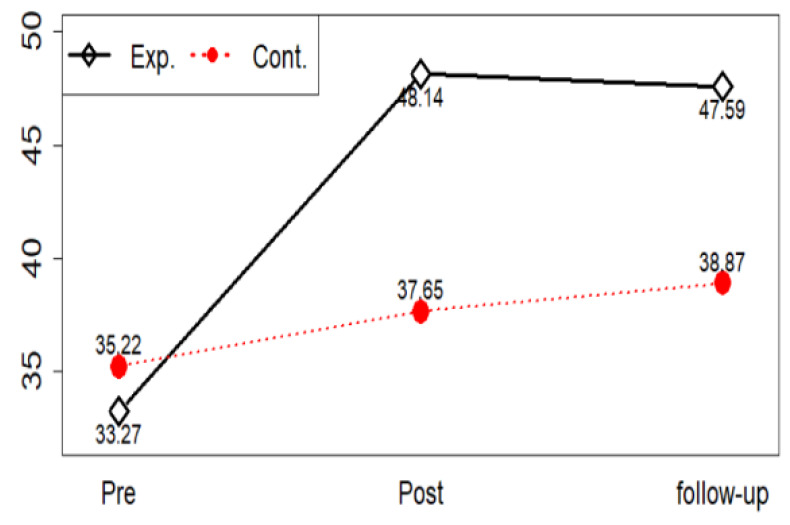
Problem-focused coping style profile plots.

**Figure 5 ijerph-17-04004-f005:**
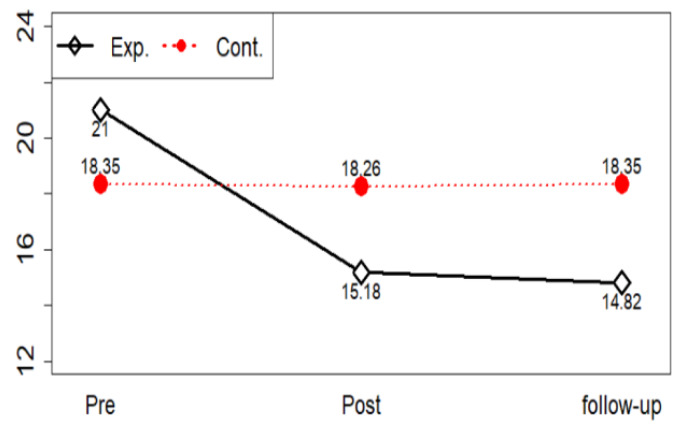
Emotion-focused coping style profile plots.

**Table 1 ijerph-17-04004-t001:** Workplace Violence Prevention Program.

Session	Themes	Contents
1	Program orientation Intimacy and trust through self-introductions Need to prevent and cope with hospital violence	Warm up (20 min) Orientation Activity (80 min) Preliminary Survey, video on hospital violence Violence Presentation and countermeasures Wrap up (20 min) Sharing impressions, self-reflection
2	Understanding therapeutic and non-communicative communication	Warm up (20 min) Break past session Activity (80 min) Therapeutic communication, video on non-therapeutic communication Presenting role-plays by team Wrap up (20 min) Sharing impressions, self-reflection
3	Listen and speak well	Warm up (20 min) Program introduction and schedule guide Activity (80 min) Listening training, well speaking training Wrap up (20 min) Sharing impressions, self-reflection
4	Communication self-check	Warm up (20 min) Program introduction and schedule Activity (80 min) Communication Wrap up (20 min) Sharing impressions, self-reflection
5	Verbal violence prevention and response	Warm up (20 min) Program introduction and schedule Activity (80 min) Production of violence scenarios, role play announcement Wrap up (20 min) Sharing impressions, self-reflection
6	Prevention of and coping with physical violence	Warm up (20 min) Program introduction and schedule YouTube video on physical violence Activity (80 min) Scenario presentation as a role play Wrap up (20 min) Sharing impressions, self-reflection
7	Prevention and coping by type of violence	Warm up (20 min) Program introduction and schedule Activity (80 min) Scenario presentation as a role play Wrap up (20 min) Sharing impressions, self-reflection
8	Evaluate program effectiveness	Warm up (20 min) Share changes after attending the entire program Activity (80 min) Summarizing program Ex-post survey, education satisfaction survey (response), evaluation Wrap up (20 min) Sharing impressions, self-reflection
Follow-up session	Violence prevention and coping through simulation	Warm up (20 min) Simulation situation setup Activity (120 min) Module: Emergency Room Violence Simulation: 2 h (action) Wrap up (20 min) Sharing impressions, self-reflection

**Table 2 ijerph-17-04004-t002:** Homogeneity test of general characteristics (*n* = 45).

Characteristics	Division	All (*n* = 45)	Exp. (*n* = 22)	Cont. (*n* = 23)	x^2^	*p*-Value
		*n* (%)	*n* (%)	*n* (%)		
Nursing major satisfaction	Very good	12 (26.7)	7 (58.3)	5 (41.7)	1.13 *	0.856
	Good	23 (51.1)	11 (47.8)	12 (52.2)		
	Moderate	8 (17.8)	3 (37.5)	5 (62.5)		
	Dissatisfaction	2 (4.4)	1 (50)	1 (50)		
Clinical practice satisfaction	Very good	8 (17.8)	4 (50)	4 (50)	3.33 *	0.380
	Good	22 (48.9)	8 (36.4)	14 (63.6)		
	Moderate	12 (26.7)	8 (66.7)	4 (33.3)		
	Dissatisfaction	3 (6.7)	2 (66.7)	1 (33.3)		
Violence situation experience period	Day	31 (68.9)	16 (51.6)	15 (48.4)	0.29	0.749
	Evening	14 (31.1)	6 (42.9)	8 (57.1)		
Security officer	Yes	13 (28.9)	4 (30.8)	9 (69.2)	2.40	0.189
	No	32 (71.1)	18 (56.3)	14 (43.8)		
Education necessity	Moderate	2 (4.4)	2 (100)	0 (0)	2.26 *	0.364
	Mild need	15 (33.3)	8 (53.3)	7 (46.7)		
	Very need	28 (62.2)	12 (42.9)	16 (57.1)		
Theoretical grades	High	11 (24.4)	9 (81.8)	2 (18.2)	6.30 *	0.054
	Moderate	25 (55.6)	10 (40)	15 (60)		
	Low	9 (20.0)	3 (33.3)	6 (66.7)		
Practicum grades	High	40 (88.9)	21 (52.5)	19 (47.5)	4.72 *	0.109
	Moderate	4 (8.9)	0 (0)	4 (100)		
	Low	1 (2.2)	1 (100)	0 (0)		

Exp. = experimental group; Cont. = control group; G = group; T = time; * Fisher’s exact test.

**Table 3 ijerph-17-04004-t003:** Test of normality of dependent variables (*n* = 45).

Variable	Exp.(*n* = 22)	*p*-Value	Cont.(*n* = 23)	*p*-Value
	Kolmogorov-Smirnov		Kolmogorov-Smirnov	
Communication self-efficacy	0.16	0.102	0.09	0.200
Problem-focused coping style	0.16	0.127	0.10	0.200
Emotion-focused coping style	0.09	0.200	0.16	0.109
Coping with violence	0.27	<0.001	0.17	0.060

**Table 4 ijerph-17-04004-t004:** Box’s test of equality of covariance matrix (*n* = 45).

Variable	Box’s M	*p*-Value
Communication self-efficacy	0.572	0.753
Problem-focused coping style	1.427	0.200
Emotion-focused coping style	0.827	0.549

**Table 5 ijerph-17-04004-t005:** Mauchly’s Test of Sphericity (*n* = 45).

Mauchly’s Test of Sphericity.							
Within Subjects Effect	Mauchly’s W	Approx. Chi-Square	Df	Sig.	Epsilon		
					Greenhouse-Geisser	Huynh-Feldt	Lower-Bound
Communication self-efficacy	0.887	5.014	2	0.082	0.899	0.957	0.500
Problem-focused coping style	0.936	2.758	2	0.252	0.940	1.000	0.500
Emotion-focused coping style	0.958	1.814	2	0.404	0.959	1.000	0.500

**Table 6 ijerph-17-04004-t006:** Repeated-measures ANOVA (*n* = 45).

Variable	Groups	Pre	Post 1	Post 2	Sources	F(*p*)
		Mean ± SD	Mean ± SD	Mean ± SD		
Communication self-efficacy	Exp.(*n* = 22)	136.14 ± 10.54	142.73 ± 16.02	138.23 ± 8.01	G	0.03 (0.845)
					T	1.08 (0.343)
	Cont.(*n* = 23)	137.87 ± 11.4	138.57 ± 17.92	139.17 ± 9.72	G*T	0.83 (0.437)
Problem-focused coping style	Exp.(*n* = 22)	33.27 ± 6.44	48.14 ± 3.62	47.59 ± 4.71	G	31.61 (<0.001)
	Cont.(*n* = 23)	35.22 ± 6.15	37.65 ± 5.42	38.87 ± 4.93	T	47.66 (<0.001)
					G*T	20.77 (<0.001)
Emotion-focused coping style	Exp.(*n* = 22)	21.00 ± 3.59	15.18 ± 4.57	14.82 ± 3.29	G	2.08 (0.156)
					T	12.35 (<0.001)
	Cont.(*n* = 23)	18.35 ± 4.67	18.26 ± 4.93	18.35 ± 3.11	G*T	12.03 (<0.001)

Exp. = experimental group; Cont. = control group; G = group; T = time; SD = standard deviation.

**Table 7 ijerph-17-04004-t007:** Observation and evaluation score for coping with violence of experimental and control groups (N = 45).

Group	Mean ± SD	U *	*p*-Value
Exp.(*n* = 22)	10.27 ± 2.10	70	<0.001
Cont.(*n* = 23)	5.70 ± 2.10		

Exp. = experimental group; Cont. = control group; G = group; T = time; * Mann–Whitney U-test; SD = standard deviation.

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
