# Peer review of "The Development and Effectiveness of a Clinical Training Violence Prevention Program for Nursing Students"

_ijerph, 2020, doi:10.3390/ijerph17114004_

Round 1

Reviewer 1 Report

It seems to me a very interesting study, before a very prevalent problem within health proffesionals and exportable to any country.

Author Response

Dear Reviewer 1:

I wish to resubmit a research article for publication in the International Journal of Environmental Research and Public Health, titled “The Development and Effectiveness of a Clinical Training Violence Prevention Program for Nursing Students.” The paper was coauthored by Prof. Dr. Jeong.

According to the comments, I made the following revisions to the manuscript.

Following reviewer 2’s comment, the proper design of the design and the characteristics of the research participants were explained. We described the instruments including how they are evaluated and the range of scores. The procedure was also described in depth. We explained when the χ2 test and the exact Fisher’s test were performed to determine homogeneity. We explained the assumptions for the repeated measures ANOVA properly in this study and also concluded based on multiple comparison tests.

Following reviewer 3’s comment, the manuscript was proofread for native use of English. We tried to include sufficient details of background and all relevant references in the introduction section. We presented conclusions and limitations of the study according to the results.

We are very grateful that our paper has been improved with the all the reviewers’ appropriate and insightful comments.

We look forward to hearing from you.

Reviewer 2 Report

  • Introduction. The authors adequately substantiate the need for research on coping with violence in the context of student nurses. Similarly, the importance of including training in coping strategies for violence in student education seems clear.
  • Study Design. Authors indicate that the research is quasi-experimental research using a non-equivalent control group pretest-posttest design. However, it appears that the control and experimental groups were not previously formed. In the section participants the authors say “Students selected as participants were randomly assigned to the 118 experimental group (N=22) and the control group (N=23), according to the training cycle”. Was the design quasi-experimental or experimental? It is not clear. 
  • Participants should be better described, based on characteristics such as age or sex as these are variables that have shown to be related to coping styles in violence.
  • The description of the instruments does not include how they are evaluated, range of scores. This description is important for data analysis and interpretation of results

Data collection:

  • The procedure should be better described. From the description of the authors, it seems that both the evaluator and the researchers who implemented the intervention knew the research objectives. This should be clarified.
  • The text indicates “three professors performed an observational assessment of the participants’ ability to cope with violence while running the simulation” Did the three teachers observe all the participants? Did each teacher observe a number of students? Three teachers should have observed all participants and made a measurement of interrater reliability such us percent agreement or Kappa. It should also be made clear whether the professors knew the objectives of the research. Knowledge of the objectives could invalidate the evaluation, since objectivity is difficult.
  • Students in the control group were given a 120-minute lecture in each session. What were the contents of the lectures? This description should be included in the text.

Data analysis and Results

  • The authors should explain when the X2 test and the exact Fisher test were performed to determine homogeneity. Especially, considering the small number of participants at some levels of the variables analyzed.
  • Repeated ANOVA measurements were made. The results of the homogeneity test are indicated, but not the other assumptions required by ANOVA. The assumptions for the repeated measures ANOVA are: a) Independent and Identical distributed variables (independent observations); b) Normality (test variables follow a multivariate normal distribution in the population); and c) Sphericity (variations of all difference scores between test variables should be equal in the population). Considering the characteristics of the participants, it is difficult for the assumptions for the ANOVA to be fulfilled. Therefore, an ANOVA would be incorrect. An alternative could be the Linear Mixed Model. The Linear Mixed Model is just an extension of the general linear model in which the linear predictor contains random effects in addition to the usual fixed effects.
  • The title of table 3 is not correct.Table 3 shows the comparison between groups (Experimental and Control) and evaluation times (Pre-Post1-Post2). Also, the organization of the data in the table is confusing by including in the group line (control or experimental) the main effects (G/T) and the interaction effects of the ANOVA (G*T).
  • The authors indicate that the experimental group experienced more statistically significant effects in terms of the problem-focused coping style in the pretest, posttest, and follow-up assessment than the control group. This conclusion should be based on multiple comparison tests. However, the authors do not report that they have done so and neither do the results of the comparison tests between the different times (Pretest- Post1/Post1-Post2)

Author Response

26 May 2020

Dear Reviwer 2

I wish to resubmit a research article for publication in the International Journal of Environmental Research and Public Health, titled “The Development and Effectiveness of a Clinical Training Violence Prevention Program for Nursing Students.” The paper was coauthored by Prof. Dr. Jeong.

According to the comments, I made the following revisions to the manuscript.

Following reviewer 2’s comment, the proper design of the design and the characteristics of the research participants were explained. We described the instruments including how they are evaluated and the range of scores. The procedure was also described in depth. We explained when the χ2 test and the exact Fisher’s test were performed to determine homogeneity. We explained the assumptions for the repeated measures ANOVA properly in this study and also concluded based on multiple comparison tests.

Following reviewer 3’s comment, the manuscript was proofread for native use of English. We tried to include sufficient details of background and all relevant references in the introduction section. We presented conclusions and limitations of the study according to the results.

We are very grateful that our paper has been improved with the all the reviewers’ appropriate and insightful comments.

We look forward to hearing from you.

Sincerely,

Kyunghee Lee

Reviewer 3 Report

This article was easy to read. What an interesting research project. The article was well organized and included standard article components. In-text citations require some refining. In data collection section, last sentence needs some clarification. Good introduction. Purpose of study were stated along with hypothesis. The image of research program development steps was clearly illustrated conceptual model. There was a good description of instruments and methodology. The results section adequately described findings. After reading the findings sections, what are the implications and limitations of this study? In addition, conclusion paragraph needs further development.

Author Response

Dear Reviewer 3:

I wish to resubmit a research article for publication in the International Journal of Environmental Research and Public Health, titled “The Development and Effectiveness of a Clinical Training Violence Prevention Program for Nursing Students.” The paper was coauthored by Prof. Dr. Jeong.

According to the comments, I made the following revisions to the manuscript.

Following reviewer 2’s comment, the proper design of the design and the characteristics of the research participants were explained. We described the instruments including how they are evaluated and the range of scores. The procedure was also described in depth. We explained when the χ2 test and the exact Fisher’s test were performed to determine homogeneity. We explained the assumptions for the repeated measures ANOVA properly in this study and also concluded based on multiple comparison tests.

Following reviewer 3’s comment, the manuscript was proofread for native use of English. We tried to include sufficient details of background and all relevant references in the introduction section. We presented conclusions and limitations of the study according to the results.

We are very grateful that our paper has been improved with the all the reviewers’ appropriate and insightful comments.

We look forward to hearing from you.

Sincerely,

Kyunghee Lee, DNS, RN

Round 2

Reviewer 2 Report

The explanations provided to clarify some aspects have been adequate. However, it is not clear to me that the research was quasi-experimental with a non-equivalent control group, as both were randomly assigned to the control and experimental groups.